# Combined Effects of Age and Comorbidities on Electrocardiographic Parameters in a Large Non-Selected Population

**DOI:** 10.3390/jcm11133737

**Published:** 2022-06-28

**Authors:** Paolo Giovanardi, Cecilia Vernia, Enrico Tincani, Claudio Giberti, Federico Silipo, Andrea Fabbo

**Affiliations:** 1Cardiology Service, Department of Primary Care, Health Authority and Services of Modena, 41124 Modena, Italy; 2Cardiology Unit, Ospedale S. Agostino–Estense, Azienda Ospedaliero-Universitaria Modena, 41126 Baggiovara, Italy; 3Department of Physics, Informatic and Mathematics, University of Modena and Reggio Emilia, 41125 Modena, Italy; cecilia.vernia@unimore.it; 4Internal Medicine Division, Ospedale S. Agostino–Estense, Azienda Ospedaliero-Universitaria Modena, 41126 Baggiovara, Italy; tincani.enrico@aou.mo.it; 5Department of Sciences and Methods for Engineering, University of Modena and Reggio Emilia, 42122 Reggio Emilia, Italy; claudio.giberti@unimore.it; 6Department of Clinical Engineering, Health Authority and Services and Azienda Ospedaliero-Universitaria Modena, 41124 Modena, Italy; f.silipo@ausl.mo.it; 7Geriatric Service—Cognitive Disorders and Dementia, Department of Primary Care, Health Authority and Services of Modena, 41124 Modena, Italy; a.fabbo@ausl.mo.it

**Keywords:** electrocardiogram, ECG, electrocardiography, Z scores, ageing, reference values, QRS-T angle, PR interval, QT corrected, heart axis

## Abstract

Background: Previous studies have evaluated average electrocardiographic (ECG) values in healthy subjects or specific subpopulations. However, none have evaluated ECG average values in not selected populations, so we examined ECG changes with respect to age and sex in a large primary population. Methods: From digitized ECG stored from 2008 to 2021 in the Modena province, 130,471 patients were enrolled. Heart rate, P, QRS and T wave axis, P, QRS and T wave duration, PR interval, QTc, and frontal QRS-T angle were evaluated. Results: All ECG parameters showed a dependence on age, but only some of them with a straight-line correlation: QRS axis (*p* < 0.001, R^2^ = 0.991, r = 0.996), PR interval (*p* < 0.001, R^2^ = 0.978, r = 0.989), QTc (*p* < 0.001, R^2^ = 0.935, r = 0.967), and, in over 51.5 years old, QRS-T angle (*p* < 0.001, R^2^ = 0.979, r = 0.956). Differences between females and males and in different clinical settings were observed. Conclusions: ECG changes with ageing are explainable by intrinsic modifications of the heart and thorax and with the appearance of cardiovascular diseases and comorbidities. Age-related reference values were computed and applicable in clinical practice. Significant deviations from mean values and from Z-scores should be investigated.

## 1. Introduction

The Electrocardiogram (ECG) is an inexpensive and non-invasive diagnostic technique used worldwide for the management of acute and chronic cardiovascular and not cardiovascular diseases; ECG is considered an outdated investigation but nowadays many clinical decisions are still based on ECG interpretation.

Many years ago, normal ECG values were established [1], and previous studies have evaluated average ECG values in specific subpopulations, such as healthy subjects [2], athletes [3], females [4], young [5] and elderly people [6], showing that ECG values should be corrected for age [7] and ethnicity [8,9]. To our knowledge, none of the previous studies have evaluated ECG average values in a large non-selected population with cardiovascular diseases and with comorbidities.

Moreover, the use of digitized programs for automated ECG interpretation has demonstrated a good accuracy and a favorable cost–benefit analysis.

Therefore, this cross-sectional observational study was designed to analyze the effect of age and sex on an extensive set of ECG parameters (heart axis, waves duration, times of conduction and repolarization, frontal QRS-T angle) in a large contemporary primary population.

## 2. Material and Methods

### 2.1. Study Design

A retrospective review of digitized ECGs recorded was made within the province of Modena from January 2008 to September 2021, together with retrospective clinical data collection. The study was performed in accordance with the Ethical Standards of the 1975 Helsinki Declaration revised in 2013 and was approved by the local Ethic Committee of Modena (AVEN) with protocol number 2605/2021, date of approval 21 September 2021.

### 2.2. Study Population

The province of Modena, located in central Emilia-Romagna, Italy, has a population of 702,549 people (www.provincia.modena.it, accessed on 30 September 2021) spread across 47 municipalities.

National health general practitioners refer their patients to national health system core facilities for clinical tests; ECGs are recorded in Emergency Departments, surgical and medical Hospital Units, and in- and out-of-hospital Ambulatories.

All patients with a digitized ECG archived in any facility within the Province of Modena were eligible.

### 2.3. ECG Recording and Inclusion–Exclusion Criteria

ECGs were recorded at rest in the supine position using a standard 12-lead tracing at 25 mm/s speed, 10 mm/mV amplitude, with a sampling rate of at least 500 Hz and were archived into a “MUSE^®^” electronic archive (GE Marquette Healthcare, Milwaukee, WI, USA).

Automated analysis was performed through a digitized multi-channel computer-assisted program (GE 12SL ECG Analysis), which uses validated algorithms for ECG parameters measurement. ECG diagnoses were confirmed by trained cardiologists.

ECGs were discarded when incomplete or had a bad signal quality, waveform recognition errors, or electrode interchanges. ECGs were also discarded in patients with pacemaker or implantable cardioverter-defibrillator, with complete or incomplete left or right bundle-branch block, with atrial fibrillation or atrial flutter, with supraventricular tachycardia, Wolff–Parkinson–White syndrome and second- or third-degree atrioventricular block. ECGs of pediatric patients (up to 15 years of age) and of very old people (more than 90 years of age) were also excluded.

In the case of patients with multiple ECGs archived in the dataset, only the first one was included. ECGs were then separated into three groups according to the facility where they were recorded.

### 2.4. Clinical Data

For the enrolled patients, age, sex, demographic data, cardiovascular risk factors, and comorbidities were retrospectively collected from in- and out-of-hospital databases. Diabetes was defined as a group of metabolic disorders characterized by hyperglycemia resulting from defects in insulin secretion, insulin action, or both. Systemic arterial hypertension was defined as the increase of systolic blood pressure equal to or above 140 mmHg and/or a diastolic blood pressure equal to or above 90 mmHg. Dyslipidemia was defined as a disorder in lipoprotein metabolism resulting in elevation of the blood concentration of cholesterol and/or triglycerides and tobacco smoke as the active exposure to tobacco products. Heart failure was defined as a syndrome with symptoms and or signs caused by structural and/or functional cardiac abnormalities. Coronary artery diseases were defined as a group of diseases characterized by the reduction of blood flow to the heart muscle because of a partial or complete blockage of the coronary arteries. Stroke was defined as a neurological deficit of cerebrovascular cause. Chronic obstructive pulmonary disease was defined as a chronic inflammatory lung disease that causes obstructed airflow. Dementia was defined as the chronic deterioration in cognitive function not expected from the usual consequences of biological aging. Cancer was defined as a large group of diseases starting in any organ or tissue of the body when abnormal cells grow uncontrollably and can invade other parts of the body. Chronic kidney disease was defined as a kidney damage or glomerular filtration rate <60 mL/min/1.73 m^2^ for more than three months.

### 2.5. ECG Parameters

The following parameters were extracted from the MUSE^®^ electronic archive [10,11]:-Heart rate.-P, QRS, and T wave axes were recorded from peak amplitudes in the extremity leads.-P and T wave durations were calculated through the mean of the 12 ECG leads.-PR interval was defined as the time between the beginning of atrial depolarization to the onset of ventricular depolarization and was measured through the mean of the 12 ECG leads. Prolonged PR interval, or first-degree atrioventricular block, was defined by an interval greater than 212 milliseconds (ms) [10].-QRS complex duration was established through the mean of the 12 ECG leads.-QT interval was defined as the time between the beginning of ventricular depolarization and the end of ventricular repolarization measuring the mean of the 12 leads. QT corrected (QTc) was established through Bazett’s correction and a prolonged QTc was defined by an interval greater than 457 ms [10].-Frontal QRS-T angle was calculated as the absolute difference in value between the frontal plane QRS and T wave axes. If the difference between the QRS and T wave axes was greater than 180 degrees (deg) the resultant QRS-T angle would be calculated as 360 deg minus the absolute angle to obtain a value between 0 deg and 180 deg [12,13].

ECG data were transferred into an Excel file and statistical analysis was performed.

### 2.6. Statistical Analysis

Continuous variables are displayed as mean ± standard deviation, while categorical data are displayed as frequencies. A two-tailed *p*-value ≤ 0.05 was considered statistically significant, with a 95% confidence interval.

The mean values ± standard deviation of the ECG parameters together with −2 and +2 Z-scores in the whole population and in the three groups (Emergency Departments, Hospital Units, and in- and out-of-hospital Ambulatories) were computed. The ANOVA test to check the hypothesis that the three groups came from populations with the same mean against the alternative hypothesis that the population means were not all the same was performed. *T*-test (or Wilcoxon test in the case of not-normal data) was used to check the same null hypothesis for pairs of subgroup data.

The difference of means between males and females in the whole population was analyzed through the *t*-test.

To investigate the null hypothesis of independence of ECG parameters from age, a chi-square test was performed. Regression analysis was used to assess the linear dependence of ECG parameters on age and the coefficient of determination R^2^ was used to determine the best linear fit. Pearson’s correlation coefficient (r) values were also given.

For the significant parameters at linear regression analysis, mean values and −2 and +2 Z-scores (corresponding to 2.3rd and 97.7th percentiles) were tabulated according to 15 age categories (5 years each, from 15 to 90 years old). The moving mean values over a sliding 5-year window, together with the standard deviations over the same moving windows, were calculated. Analyses were performed with MATLAB (R 2021 b).

## 3. Results

### 3.1. Population Data

From January 2008 to September 2021, 309,405 ECGs were archived in the MUSE^®^ electronic archive in the national health system facilities of Modena. Of these, after exclusions, 130,471 patients were enrolled. The Emergency Department group was composed of 29,560 patients, the Hospital Units group of 85,239 patients, and the Ambulatory group of 15,672 patients (baseline characteristics of the study population and exclusion criteria are shown in Table 1). The database comprised 1,565,652 data points.

The prevalence of cardiovascular risk factors in the study population according to age (15–54 years vs. 55–90 years) were: diabetes 3.3% vs. 16.3%, hypertension 9.6% vs. 38.2%, dyslipidemia 17.6% vs. 36.4%, and tobacco smoke 18.1% vs. 3.9%, respectively.

The prevalence of cardiovascular diseases and comorbidities in the study population according to age (15–54 years vs. 55–90 years) were: cardiovascular diseases 3.3% vs. 18.2%, cerebrovascular diseases 0.9% vs. 9.4%, chronic obstructive pulmonary diseases 7.5% vs. 15.9%, dementia 0.3% vs. 5.1%, cancer 4.9% vs. 19%, and chronic kidney disease 0.8% vs. 8.2%, respectively.

Figure 1 shows the age distribution of the enrolled patients, while Table 2 shows average ECG values and Z-scores in the whole population and in the Emergency Department, Hospital Units, and in- and out-of-hospital Ambulatories groups.

### 3.2. ECG Analysis

The effects of age, sex, and of the different facilities on the considered ECG parameters were:-Heart rate significantly increased with ageing, without a linear correlation (*p* < 0.001, R^2^ = 0.06, r= −0.246, Figure 2a). Females had greater heart rate than males (74.6 ± 14.3 beats per minute (bpm) vs. 70.9 ± 14.6 bpm, respectively, *p* < 0.001, Figure 3a). Heart rate was greater in the Emergency Department group (Table 2).

-P wave axis slightly but significantly turned to the right, without a linear correlation (*p* < 0.001, R^2^ = 0.692, r = 0.832, Figure 2b). Females had greater values than males (50.8 ± 22.2 deg vs. 49.6 ± 22.6 deg, respectively, *p* = 0.014, Figure 3b). P wave axis turned to the right mainly in the Emergency Department facility (Table 2).-QRS axis with a straight-line correlation turned to the left (*p* < 0.001, R^2^ = 0.991, r= −0.996, Figure 2c). Differences between females and males were not statistically significant (30.8 ± 37.1 deg vs. 24.5 ± 41.3 deg, respectively, *p* = 0.177, Figure 3c). A greater shift to the left was evident in the Hospital Units group (Table 2).-T wave axis significantly turned to the right (*p* < 0.001, R^2^ = 0.419, r = 0.648, Figure 2d), but without a linear correlation. No significant differences between females and males were observed (42.6 ± 31.1 deg vs. 41.2 ± 32.6 deg, respectively, *p* = 0.819, Figure 3d). T wave shifted to the right mainly in the Emergency Department group (Table 2).-P wave duration increased with ageing without a linear correlation (*p* < 0.001, R^2^ = 0.051, r = −0.227, Figure 2e). *p* duration was lower in females with respect to males (95.9 ± 19.1 ms vs. 100.9 ± 19.8 ms, respectively, *p* < 0.001, Figure 3e). Greater P waves were observed in the Emergency Department group (Table 2).-QRS duration significantly increased with ageing (*p* = 0.001, R^2^ = 0.608, r = 0.78, Figure 2f), but without a linear correlation. QRS was shorter in females than in males along all ages (84.9 ± 11.4 ms vs. 94.3 ± 13.5 ms, respectively, *p* < 0.001, Figure 3f). Greater QRS values were recorded in the Emergency Department group (Table 2).-T wave duration significantly changed with ageing, without a linear correlation (*p* < 0.001, R^2^ = 0.302, r = 0.55, Figure 2g). T wave duration was greater in females with respect to males (192.6 ± 43.9 ms vs. 185.5 ± 35.0 ms, respectively, *p* < 0.001, Figure 3g). Greater T waves were observed in the Ambulatory group (Table 2).-PR interval increased with a straight-line correlation with ageing (*p* < 0.001, R^2^ = 0.978, r = 0.989, Figure 2h). PR was shorter in females with respect to males along all ages (151.1 ± 25.9 ms vs. 160.5 ± 28.6 ms, respectively, *p* < 0.001, Figure 3h). No significant differences among the three groups were observed (Table 2).-QTc increased with a straight-line correlation (*p* < 0.001, R^2^ = 0.935, r = 0.967, Figure 2i). Females had longer QTc values with respect to males (434.6 ± 27.7 ms vs. 425.4 ± 29.5 ms, respectively, *p* < 0.001, Figure 3i) but with increasing age the differences became null. QTc values were greater in the Emergency Department group (Table 2).-Frontal QRS-T angle increased without a linear correlation (*p* < 0.001, R^2^ = 0.717, r = 0.847, Figure 4a) but when the analysis was performed including only patients older than 51.5 years, QRS-T angle revealed a straight-line correlation with ageing (*p* < 0.001, R^2^ = 0.979, r = 0.956, Figure 4b). No significant differences between females and males (30.8 ± 30.6 deg vs. 33.5 ± 33.1 deg, respectively, *p* = 0.203, Figure 4c) and between the three groups were observed (Table 2). The proportion of patients with a QRS-T angle greater than 90 deg rapidly increased among patients older than 51.5 years (Figure 4d).

Table 3 and Table 4. show, respectively, the mean values ± standard deviation and Z-scores −2 and +2 (computed in 15 age categories of 5 years each) of parameters with a linear correlation with ageing and of frontal QRS-T angle in the whole population, in males and females.

## 4. Discussion

In this large non-selected population, ageing and comorbidities influenced the considered ECG parameters, but a straight-line correlation was found only for QRS wave axis, PR, and QTc intervals. Secondarily, significant differences between females and males were found. Lastly, different mean values for some ECG parameters were registered mainly in the Emergency Department group.

Ageing produces structural and functional changes in the cardiovascular system, which involves the appearance of cardiovascular diseases and comorbidities [14]. The consequences of these conditions are the appearance of vascular stiffness, fibrosis, hypertrophy, and the involution of muscular tissue, valves, and arteries [15]. Moreover, with increasing age, heart position into the thorax changes, thoracic impedance grows, the use of heart conduction-modifying drugs increases, and the exposure to environmental factors rises [16,17].

Jorgensen and coll. already demonstrated that ECG changes with ageing were associated with the appearance of cardiovascular risk factors [18], while in a large population of Latinos Silva and coll. observed a greater prevalence of ECG abnormalities in old males [19].

We found that the straight-line changing parameters were an expression of the leftward deviation of the ventricular axis (QRS axis), of the slowdown conduction between atria and ventricles (PR interval), and of the prolongation of ventricular repolarization (QTc interval). Linear correlations were not found for P, QRS, and T waves duration and for P and T waves axes.

In healthy population studies, Van der Ende and coll. previously observed that increasing age was associated with a linear increase of PR and QT intervals and with a weak increase of P and QRS waves duration [2]. Rijnbeek and coll. in a smaller population observed the same results, excluding the stability of QRS duration [14], while Palhares and coll. in a large healthy Brazilian population evaluated mean ECG values also at extreme ages, observing smaller changes with increasing age [9].

We claim that our results are similar, but more applicable to clinical practice, having been obtained in a large non-selected population with cardiovascular risk factors and acute and chronic diseases that had been excluded from previous healthy population studies.

Heart rate: the increase in heart rate is a well-known risk factor for mortality [20]. In this study, it was higher in young patients, mildly decreased in the middle-aged population, and then increased again in old people, and overall, was higher in females. Previous healthy population studies showed fewer fluctuations in heart rate with increasing age [2,9,10,14]. We claim that in our study heart rate changes in old subjects were mainly attributable to the enrolled patients with acute diseases.

P, QRS, and T wave axes: changes in the cardiac axis have been associated with an increased risk of death [21]. In this study, P and T wave axes weakly turned to the right while QRS axis turned to the left with a straight-line correlation. Previous healthy population studies have observed smaller changes in the heart axis with increasing age [2,9,10,14]. Regardless of the QRS axis, our results could be explained by the appearance of hypertension and diabetes and especially by changes in heart position and in thoracic impedance.

P, QRS, and T wave duration: in this study mean P and T wave duration had small changes with increasing age, while QRS duration slightly increased without a linear correlation. P and QRS duration were greater in males, while T duration was greater in females. Additionally, previous studies have observed the same results [2,9,10,14], and our work confirmed that increasing age was not strictly associated with the increase in times of contraction.

PR interval: PR duration has been associated with the appearance of atrial fibrillation and with cardiovascular death [22]. Like previous studies, our work confirmed that the PR interval—greater in males—increased with a straight-line correlation in the entire population and in the three groups.

QTc interval: QTc is a globally utilized parameter influenced by many factors, associated with an increased arrhythmic risk, and calculable with various methods (Framingham, Hodges, Fredericia, Bazett, Rautahariu) [23]. Like previous studies, we observed a linear increase of QTc with increasing age utilizing Bazett’s correction but differences between males and females became null with ageing. This behavior could be attributable to sexual hormones: in males, QTc is related to testosterone levels, causing a hormone effect on cardiac ion channels [24,25]. QTc assessment before and during the use of heart conduction-modifying drugs, for their association with an increased risk of sudden cardiac death, is strongly recommended [26].

Frontal QRS-T angle: QRS-T angle could be determined through vectorcardiography (spatial QRS-T angle) and through standard ECG recording (frontal QRS-T angle) [27]; it represents the disjunction between left ventricular depolarization and repolarization and is a strong predictor of cardiovascular death and of coronary artery diseases [28,29].

Aro and coll. showed that a frontal QRS-T angle greater than 100 deg was associated with a high risk of sudden cardiac death [30]. Except for the short report of Marcolino and coll. in a population of Latinos [31], to our knowledge, previous studies have not investigated QRS-T angle changes with ageing.

In this study, the QRS-T angle remained stable until middle age, and then rapidly increased with a straight-line correlation. QRS-T angle better reflected the appearance of cardiovascular diseases and comorbidities than the changes in heart position and thoracic impedance. The evidence of an abnormal QRS-T angle especially in young and middle-aged subjects should be explained and the underlying presence of coronary artery diseases be excluded.

Most of the considered parameters have a clinical significance and a prognostic role with respect to the appearance of atrial fibrillation, sudden cardiovascular death, and major cardiovascular events [32,33,34], so their precise characterization is crucial. This work demonstrated, in a large contemporary primary population, that ECG is influenced by ageing, sex, cardiovascular diseases, and comorbidities but is difficult to be defined the weight of each of these conditions.

The study provided mean and Z-score reference values for the ECG parameters with a straight-line correlation with age and sex. The detection of abnormal ECG values should be investigated by clinicians excluding the presence of cardiomyopathies, congenital and arrhythmogenic heart diseases, subtle hypertension, or ischemic heart diseases.

Moreover, clinicians should consider the effects of many drugs and of acute diseases, particularly infectious diseases, on ECG. A recent example is represented by SARS-CoV−2, which is able to cause ECG abnormalities and arrhythmias due to its ability to alter ion and especially calcium homeostasis [35,36,37].

For many years, ECG reporting has been considered a simple and subjective procedure, but a new era is coming. News ECG parameters are disposable and the use of algorithms such as “heart age” [38] or “age gap” [39] could increase the diagnostic and prognostic power. Moreover, the use of artificial intelligence can allow us to identify the presence of many conditions such as left ventricular systolic dysfunction, coronary artery diseases, accessory pathways, aortic stenosis, and hypertrophic cardiomyopathy [40,41,42]. At present, these new technological opportunities are not widely available, but the simple use of reference values and Z-scores could give an additional value to ECG interpretation.

## 5. Limitations

This study has limitations, mainly due to its retrospective nature, such as the unknown prevalence of patients utilizing cardiotoxic drugs or heart conduction-modifying drugs, the unknown exposure to environmental factors, and the unknown prevalence of pulmonary hypertension or left ventricular hypertrophy [43]. Despite these limitations, we claim that these unknown factors did not skew the results, because the sample size is strong and the straight-line correlations started from a young age and continued up to very old age.

## 6. Conclusions

The results of our study, obtained in a large primary population, are more applicable in clinical practice than those of previous healthy population studies conducted in order to maximize the effects of age, sex, and co-pathologies on ECG.

Most of the considered ECG parameters changed with increasing age but only a few of them with a linear correlation.

ECG modifications were also influenced by sex and comorbidities and were usually greater in the Emergency Department group.

Reference values expressed by means of and Z-scores were computed for linearly changing parameters; their use could improve the diagnostic and prognostic ability of clinicians in ECG interpretation.

Significant deviations from mean values and from Z-scores should be investigated by clinicians, and the evaluation of the QRS-T angle especially in young and middle-aged patients could be improved.

The expression of ECG values as Z-scores may provide additional information. Therefore, ECG could increasingly become a prognostic tool more than a diagnostic test.

## Figures and Tables

**Figure 1 jcm-11-03737-f001:**
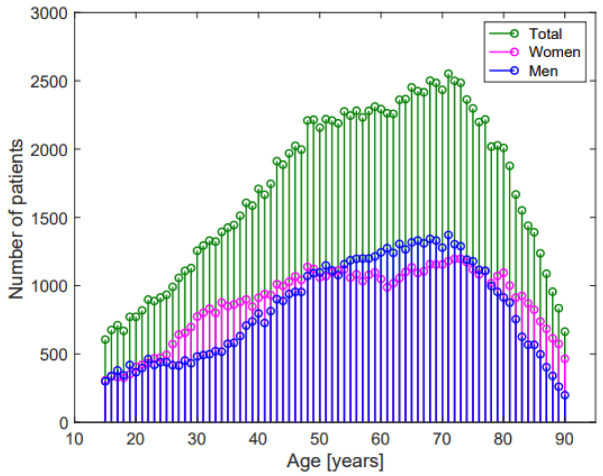
Age-distribution of the study population.

**Figure 2 jcm-11-03737-f002:**
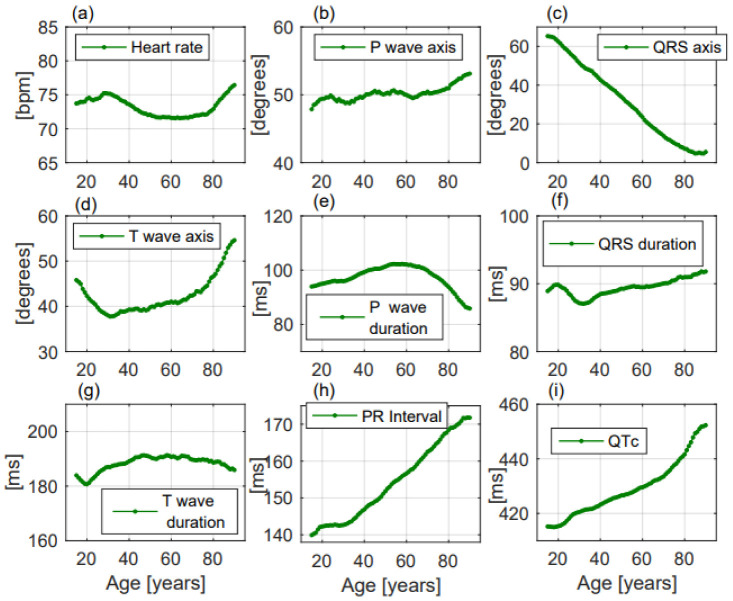
ECG changes with ageing of the considered parameters in the enrolled population. All the parameters (**a**–**i**) changed with ageing but only QRS axis (**c**), PR interval (**h**), and QTc (**i**) presented a linear change with increasing age.

**Figure 3 jcm-11-03737-f003:**
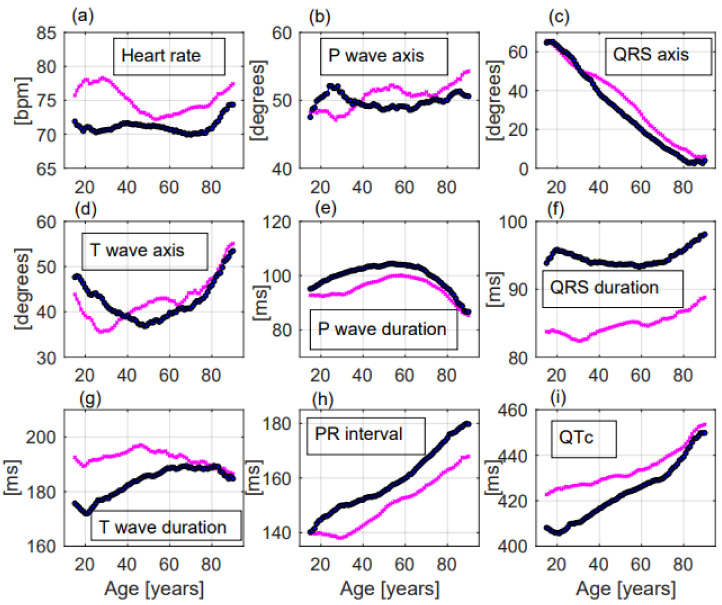
ECG changes with ageing of the considered parameters in females—and in males—Significant differences between females and males were observed for most of the ECG parameters (**a**–**i**). Especially heart rate and QTc were significantly greater in females (**a**,**i**) while QRS duration and PR interval were significantly greater in males (**f**,**h**).

**Figure 4 jcm-11-03737-f004:**
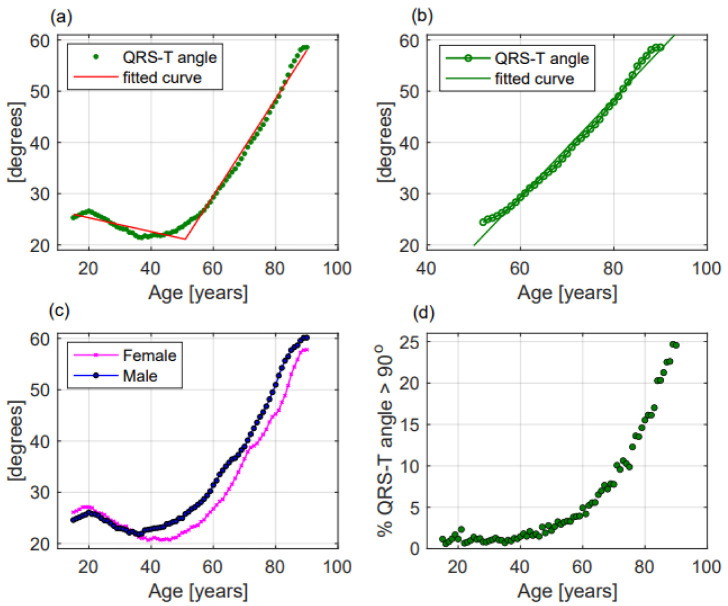
QRS-T angle modifications with ageing in the whole population (**a**), in patients older than 51.5 years (**b**), in females and males (**c**); and (**d**) percentage of patients with QRS-T angle greater than 90 deg according to age. In panels (**a**–**c**) dots represent mean values of QRS-T angle while in panel (**d**) green dots represent the percentage of patients with a QRS-T angle greater than 90 deg. In panels (**a**,**b**) lines represent the fitted curve.

**Table 1 jcm-11-03737-t001:** Baseline characteristics of the study population and exclusion criteria.

Clinical Characteristics of the Enrolled Patients:		Exclusion Criteria:	
Enrolled patients ^§^	130,471	Incorrect or incomplete ECG ^§^	2826
Men ^§^	63,261 (48.5%)	Bad signals qualities ECG ^§^	26,659
Women ^§^	67,210 (51.5%)	Young and very old people ^§^	12,902
Mean age at enrollment date (years) *	56.8 ± 19.4	Multiple ECG ^§^	89,870
Emergency Departments group ^§^	29,560	Atrial fibrillation ^§^	10,245
Men ^§^	14,658 (49.6%)	Atrial flutter ^§^	1957
Women ^§^	14,902 (50.4%)	Supraventricular tachycardia ^§^	3721
Surgical/Medical Units group ^§^	85,239	Wollf-Parkinson-White ^§^	434
Men ^§^	41,519 (48.7%)	Bundle branch block ^§^	19,660
Women ^§^	43,720 (51.3%)	Pacemaker or implantable cardioverter-defibrillator ^§^	8670
Ambulatory group ^§^	15,672	Second-degree AV block ^§^	1698
Men ^§^	7084 (45.2%)	Third-degree AV block ^§^	292
Women ^§^	8588 (54.8%)	Excluded patients ^§^	178,934

^§^ Number of patients, % percent of patients, * Mean ± standard deviation.

**Table 2 jcm-11-03737-t002:** Mean ECG values and Z-scores −2 and +2 in the study population and in the three groups.

	Study Population 130,471 Patients		Emergency Department Group 29,560 Patients		Hospital Units Group 85,239 Patients		Ambulatory Group 15,672 Patients		
	Mean ± sd *		Mean ± sd *		Mean ± sd *		Mean ± sd *		*p*-Value
	Z Scores −2	Z Scores +2	Z Scores −2	Z Scores +2	Z Scores −2	Z Scores +2	Z Scores −2	Z Scores +2	
**Heart rate** (bpm ^°^)	72.8 ± 14.5		76.9 ± 16.7		72.1 ± 13.8		69.0 ± 12.0		<0.001
	50	108	50	116	50	105	49	97	
**P wave axis** (deg ^^^)	50.2 ± 22.4		53.8 ± 22.0		49.4 ± 22.5		48.2 ± 22.1		<0.001
	−2	85	2	86	−3	85	−2	82	
**QRS axis** (deg ^^^)	27.8 ± 39.3		34.5 ± 41.0		25.0 ± 39.2		30.1 ± 35.2		0.033
	−51	89	−52	93	−51	88	−43	86	
**T wave axis** (deg ^^^)	41.9 ± 31.8		45.7 ± 32.4		41.4 ± 32.9		37.7 ± 23.0		<0.001
	−14	110	−14.6	111	−15	116	−5	77	
**P wave duration** (ms ^#^)	98.3 ± 19.6		100.0 ±19.9		98.1 ± 19.7		96.6 ± 18.0		<0.001
	52	130	52	132	52	128	54	126	
**QRS duration** (ms ^#^)	89.5 ± 13.3		90.1 ± 13.5		89.6 ± 13.5		87.7 ± 11.6		<0.001
	70	118	70	120	70	120	68	112	
**T wave duration** (ms ^#^)	189.1 ± 40.0		187.9 ± 41.3		189.1 ± 40.7		191.4 ± 32.7		<0.001
	58	246	58	250	54	248	92	238	
**PR interval** (ms ^#^)	155.7 ± 27.6		156.6 ± 28.1		156.1 ± 27.9		152.2 ± 24.8		0.302
	112	220	114	222	112	222	112	208	
**QTc interval** (ms ^#^)	430.1 ± 29.0		435.7 ± 29.0		429.8 ± 29.4		421.0 ± 23.8		<0.001
	379	492	384	496	379	494	374	468	
**Frontal QRS-T angle** (deg ^^^)	32.1 ± 31.9		32.6 ± 33.1		33.0 ± 32.6		26.4 ± 24.2		0.073
	1	131	1	135	1	135	1	94	

* mean ± standard deviation, ^°^ beats per minute, ^^^ degrees, ^#^ millisecond.

**Table 3 jcm-11-03737-t003:** Mean ECG values, computed in 15 age categories of 5 years each, of the parameters with a straight-line correlation with age and mean QRS-T angle values. For each age category, in the upper line, means ± standard deviation in the whole population; in the bottom line, means ± standard deviation in males and females.

Age (Years)	QRS Axis * (deg ^^^)		PR Interval * (ms ^#^)		QTc Interval * (ms ^#^)		QRS-T Angle * (deg ^^^)	
	Men *	Women *	Men *	Women *	Men *	Women *	Men *	Women *
**15–19**	64.9 ± 24.7		140.7 ± 20.8		415.1 ± 26.2		25.9 ± 19.4	
	65.2 ± 26.2	64.6 ± 23.0	141.7 ± 20.0	139.5 ± 21.6	406.7 ± 25.3	424.0 ± 24.4	25.1 ± 18.9	26.7 ± 19.8
**20–24**	60.1 ± 26.7		142.5 ± 20.4		416.0 ± 26.1		26.0 ± 20.0	
	61.1 ± 28.4	59.2 ± 25.0	145.9 ± 20.8	139.3 ± 19.6	406.4 ± 26.0	425.1 ± 22.9	25.7 ± 20.1	26.3 ± 19.9
**25–29**	54.9 ± 27.5		142.6 ± 20.6		419.6 ± 25.0		24.3 ± 19.1	
	57.0 ± 28.9	53.4 ± 26.4	148.4 ± 21.2	138.6 ± 19.2	409.9 ± 24.9	426.5 ± 22.6	23.9 ± 18.9	24.6 ± 19.2
**30–34**	49.5 ± 28.7		143.1 ± 21.1		421.1 ± 24.6		23.1 ± 19.3	
	48.7 ± 31.3	49.9 ± 27.1	150.2 ± 20.7	138.7 ± 20.1	411.3 ± 24.5	427.2 ± 22.6	22.6 ± 20.1	23.3 ± 18.8
**35–39**	45.9 ± 29.7		145.5 ± 21.4		422.0 ± 24.0		21.4 ± 18.8	
	43.2 ± 32.0	47.9 ± 27.8	151.3 ± 21.6	141.1 ± 20.2	414.8 ± 24.6	427.4 ± 22.0	21.9 ± 20.1	21.0 ± 17.6
**40–44**	40.9 ± 32.0		148.0 ± 20.4		424.1 ± 26.0		21.9 ± 21.5	
	36.3 ± 33.4	44.8 ± 30.2	152.9 ± 20.4	143.8 ± 19.5	417.7 ± 25.0	429.6 ± 25.5	23.0 ± 23.4	20.9 ± 19.7
**45–49**	37.0 ± 33.5		149.9 ± 21.2		425.8 ± 25.4		22.6 ± 22.8	
	32.0 ± 35.1	41.5 ± 31.3	153.9 ± 20.8	146.2 ± 20.8	420.3 ± 25.5	430.9 ± 24.1	24.2 ± 24.8	21.1 ± 20.6
**50–54**	31.6 ± 33.8		153.0 ± 21.6		427.0 ± 25.3		24.4 ± 24.6	
	26.9 ± 34.8	36.5 ± 32.1	156.0 ± 21.4	150.0 ± 21.5	423.2 ± 26.0	430.8 ± 24.1	26.2 ± 26.1	22.6 ± 22.9
**55–59**	27.2 ± 35.1		155.4 ± 22.8		428.5 ± 26.1		26.8 ± 26.9	
	23.1 ± 36.4	31.9 ± 33.0	158.5 ± 23.4	151.9 ± 21.4	424.9 ± 26.6	432.6 ± 24.9	28.7 ± 28.4	24.6 ± 24.8
**60–64**	20.9 ± 35.6		157.6 ± 24.7		430.3 ± 26.7		31.1 ± 30.3	
	17.7 ± 37.4	24.7 ± 32.9	161.3 ± 25.6	153.1 ± 22.7	427.1 ± 26.9	434.1 ± 25.9	33.5 ± 32.5	28.2 ± 27.2
**65–69**	16.4 ± 37.3		160.5 ± 26.8		432.3 ± 27.1		34.9 ± 32.8	
	13.9 ± 39.6	19.3 ± 34.2	164.9 ± 28.0	155.3 ± 24.2	429.1 ± 28.3	436.1 ± 25.1	36.7 ± 34.0	32.7 ± 31.2
**70–74**	12.1 ± 38.4		163.3 ± 29.4		435.1 ± 29.3		40.1 ± 35.5	
	9.8 ± 40.8	14.5 ± 35.5	168.2 ± 31.3	157.9 ± 26.0	431.2 ± 29.5	439.4 ± 28.5	41.3 ± 36.6	38.7 ± 34.2
**75–79**	9.2 ± 40.8		166.7 ± 32.9		439.1 ± 30.5		44.5 ± 38.7	
	7.2 ± 44.0	11.3 ± 37.2	172.2 ± 35.0	161.2 ± 29.6	436.1 ± 30.8	442.1 ± 29.9	46.8 ± 40.2	42.3 ± 37.0
**80–84**	6.2 ± 42.3		169.0 ± 35.2		444.4 ± 33.6		50.5 ± 42.4	
	2.8 ± 46.2	8.8 ± 38.9	176.1 ± 38.0	163.5 ± 31.9	442.4 ± 33.0	445.9 ± 33.9	54.2 ± 43.8	47.6 ± 41.0
**85–90**	4.9 ± 45.4		171.3 ± 37.8		451.0 ± 36.0		57.5 ± 45.9	
	3.3 ± 49.5	5.9 ± 42.9	178.5 ± 40.9	167.1 ± 35.2	448.5 ± 36.1	452.4 ± 35.8	59.3 ± 46.4	56.4 ± 45.6

* Mean ± standard deviation, ^^^ degrees, ^#^ milliseconds.

**Table 4 jcm-11-03737-t004:** Z-scores −2 and +2, computed in 15 age categories of 5 years each, of the three parameters with a straight-line correlation with age and of QRS-T angle. On each cell Z-scores −2 (on the left) and +2 (on the right). For each age category, in the upper line, −2 and + 2 Z-scores in the whole population; in the bottom line, −2 and + 2 Z-scores in males and females.

Age (Years)	QRS Axis (deg ^^^) Z Scores −2 and + 2		PR Interval (ms ^#^) Z Scores −2 and + 2		QTc Interval (ms ^#^) Z Scores −2 and + 2		QRS-T angle (deg ^^^) Z Scores −2 and + 2	
	Men	Women	Men	Women	Men	Women	Men	Women
**15–19**	3 98		104 185		364.5 466		1 77	
	−0.49 102	9 96	106 185	104 185.1	360 459	376 470	1 74	1 82
**20–24**	−5 96		108 186		366 464		1 75	
	−7 97.5	−3 94	108 190	106 184	361 455	378 467	1 73	1 75.8
**25–29**	−6 93		108 190		372 467		1 71.5	
	−7.9 96	−4 91	114 196	108 184	364 462	384 470	1 72.9	1 71
**30–34**	−14 92		108 190		373.2 469		1 74	
	−18.7 93	−9 91.5	116 197.5	104 182	364 461	383.5 471	1 77	1 71.5
**35–39**	18 91		110 192		375 469		1 69	
	−24 93	−11 89	116 198	106.6 186	369 467	385 470.7	1 77	1 65
**40–44**	−27 89		112 192		377 474		1 80.4	
	−33 89	−20 89	116 198	108 186	371 468.6	385.7 477	1 87.6	1 72.3
**45–49**	−31 88		112 196		378 477		1 88	
	−38 88	−24 88	116 200	110 192	373 473	386.7 480	1 96.2	1 79.3
**50–54**	−36 86		116 200		381 479		1 97.4	
	−40 85.9	−30 87	118 204	112 196	378 479	386 480	1 105	1 91
**55–59**	−42.5 85		116 206		382 483		1 107	
	−48 84	−33 86	118 210	114 200	378 481.6	388 484.4	1 114.6	1 97
**60–64**	−47 82		116 214		384 487		1 122	
	−51 83	−41 81	118 218	114 204	381 486	388 488	1 135	1 106.7
**65–69**	−53 82		116 222		384 491		1 132	
	−56 84	−47 80	120 230	114 210	381 492	390 490	1 138	1 124
**70–74**	−57 81		116 230		386 498		1 142	
	−61 83	−51 80	118 240	116 218	383.5 497	392.2 498	1 147.5	1 136
**75–79**	−61 84		118 244		388 510		1 153	
	−64 90	−53 81	119.4 256	116 230	384 508	393 513.3	1 154	1 151
**80–84**	−63 86		118 254		391 525		1 161	
	−67.5 90	−58 83	120 268	114 236	390 525	392 525	1 164	1 160
**85–90**	−64 99.6		116 266		392 536		2 169	
	−67 132.6	−61 90.8	116 280	116 254	390 539.3	395.2 535	2 170	2 168

^^^ degrees, ^#^ milliseconds.

## Data Availability

The data presented in this study were extracted from a “MUSE^®^” GE Marquette Healthcare electronic archive and are deposited in the Department of Clinical Engineering. Data are available on request from the corresponding author with prior authorization from the Health Authority and Services of Modena.

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
