# Peer review of "Combined Effects of Age and Comorbidities on Electrocardiographic Parameters in a Large Non-Selected Population"

_jcm, 2022, doi:10.3390/jcm11133737_

Round 1
Reviewer 1 Report
I read with interest Giovanardi et al's manuscript, in which the authors present ECG parameters on a large cohort of Italian patients.
However, in order to improve the quality of the manuscript, we recommend that the authors calculate, in addition to the mean and standard deviation the percentiles 2.3 and 97.7, which correspond to Z scores of -2 and +2, respectively. Readers can therefore be given normal values for the P wave, PR interval, QRS duration, and T wave axis.
Reviewer 2 Report
In this study the authors demonstrated the effects of age and comorbidities on ECG parametrers.
One of the major limitations of the study as the authors state, is the unknown prevalence of patients utilizing cardiotoxic drugs or heart conduction-modifying drugs, or the presence of structural or functional cardiac abnormalities (LV hypertrpophy, ventricular dysfunction-impaired LVEF, myocardial scar, pulmonary hypertension), that could significantly affect ECG parametrs (QRS axis, QRS duration, PR interval, QTc.
The addition of a section emphasizing on the clinical significance and suggesting how the clinician should interpret and further investigate specific deviations from ECG mean values, would raise the competence of this article.
Round 2
Reviewer 1 Report
The Z score computation provides a higher value for the manuscript. It can be published in its current state.
Author Response
Modena, 22-6-2022
Dear Reviewers and academic Editor,
I have corrected the manuscript following instructions of the Academic Editor and English language has been revised following reviewer's suggestions
Paolo Giovanardi

Reviewer 2 Report
The authors have adequately implemeted the required changes.
I have no further suggestions to make.
Author Response
Modena, 22-6-2022
Dear Reviewers and Academic Editor,
we have corrected the manuscript following instructions of the Academic Editor and English language has been ameliorated following reviewer's instructions
Paolo Giovanardi
